# Diversity and Distribution of Xylophagous Beetles from *Pinus thunbergii* Parl. and *Pinus massoniana* Lamb. Infected by Pine Wood Nematode

**Xu Chu** [1,2,3,†], **Qiuyu Ma** [1,2,3,†], **Meijiao Yang** [1,2,3], **Guoqiang Li** [1,2,3], **Jinyan Liu** [1,2], **Guanghong Liang** [1,2], **Songqing Wu** [1,2], **Rong Wang** [1,2], **Feiping Zhang** [1,2,*] and **Xia Hu** [1,2,3,*]

1. Forestry College, Fujian Agriculture and Forestry University, Fuzhou 350002, China; 1190429001@fafu.edu.cn (X.C.); 1180429006@fafu.edu.cn (Q.M.); 3200422082@fafu.edu.cn (M.Y.); 1200429002@fafu.edu.cn (G.L.); 000d040092@fafu.edu.cn (J.L.); guanghong.liang@fafu.edu.cn (G.L.); songqingwu@fafu.edu.cn (S.W.); wrong-insect@fafu.edu.cn (R.W.)
2. Key Laboratory of Integrated Pest Management in Ecological Forests, Fujian Agriculture and Forestry University, Fuzhou 350002, China
3. International Joint Laboratory of Forest Symbiology, Fujian Agriculture and Forestry University, Fuzhou 350002, China
* Correspondence: fpzhang@fafu.edu.cn (F.Z.); lake-autumn@fafu.edu.cn (X.H.); Tel.: +86-591-8370-0261 (F.Z.); +86-183-5006-8276 (X.H.)
† These authors contributed equally to this work.

**Abstract:** The vectors of pinewood nematode of *Bursaphelenchus xylophilus* (Steiner & Bührer, 1934) are mainly known as xylophagous beetles. Understanding the composition and distribution of these xylophagous beetles in host pine trees infected by PWN is critical to control the spread of PWN. In this study, we investigated the community structures of the xylophagous beetles in two main host trees in Fujian and Shandong, *Pinus massoniana* Lamb. and *Pinus thunbergia* Parl., in different stages of infection. All beetles were collected by dissecting the whole pine trees and then identified by their morphological characteristics and COI genes. The results showed that the diversity of xylophagous beetles was different not only between the two host pine trees but also among the different infection stages. The diversity of *P. massoniana* xylophagous beetles was significantly higher than that of *P. thunbergii*, and there were also significant differences in the different stages of PWN infection. In total, Scolytinae was the most common (53.70%), followed by Curculionidae (18.26%), Cerambycidae (16.31%), and Cleridae (6.04%). *Monochamus alternatus*, the most important vector of PWN, occupied a large niche and showed different aggregation positions during the three infection stages in both host trees. This result might be related to the resistance of bark beetles to host trees and competition with other xylophagous beetles. The community diversity of xylophagous beetles was jointly affected by both the infection stages of PWN and the spatial niche of xylophagous beetles. Knowledge of the diversity and competitive relationships among xylophagous beetles might help regulate the population dynamics of these beetles.

**Keywords:** xylophagous beetles; distribution; *Bursaphelenchus xylophilus*

## 1. Introduction

Pine wilt disease (PWD) is an internationally recognized destructive disease of pine trees caused by pinewood nematode (PWN). Infected pine trees quickly lose water and wilt to death in a short time [1,2]. Due to its strong pathogenic ability and rapid transmission, PWD is difficult to cure once pine forests are infected, which poses a great threat to China's forest ecosystem [3,4]. This disease has spread to 726 county administrative regions among the 18 provinces in China, covering an area of 1.8092 million hectares and producing a cumulative number of dead pine trees close to 19.47 million [5]. *Pinus massoniana* (Lamb, 1803) and *Pinus thunbergii* (Parl, 1868) are important afforestation trees in the north and

south of China, respectively, and are also the main host trees of *Bursaphelenchus xylophilus* (Steiner and Buhrer 1934). Therefore, to protect these pine forests, it is urgent to control the transmission of *B. xylophilus*.

Pine forests have long suffered a complex disease caused by pine wood nematode, carrier insects, other xylophagous beetles, associated fungi, bacteria, and forest abiotic factors [6,7]. The transmission of diseases mainly depends on vector insects carrying pathogen and transferring them to trees [8]. Although all reported vectors of *B. xylophilus* are currently *Monochamus* spp., *Orthotomicus erosus* (Wollaston, 1857) was previously reported to be a carrier of *Bursaphelenchus teratospicularis* (Kakuliya, 1966) and *Bursaphelenchus sexdentati* (Braasch, 2001), while *Pityogenes* sp. was reported to be a carrier of *Bursaphelenchus leoni* (Baujard, 1980) [9,10]. Therefore, in addition to *M. alternatus*, other xylophagous beetles might also be potential carriers of *B. xylophilus*.

The natural transmission of disease is tightly linked with the life history of its insect vectors and spread through the dispersal of adult beetles [11,12]. Therefore, the spatial distribution and dynamic variation of xylophagous beetles on diseased trees are important indicators of the disease [13]. Interestingly, the colonization of xylophagous beetles is often accompanied by competition [14,15], including competition for space and nutrient resources [16]. Xylophagous beetles colonizing pine trees encounter interspecific and intraspecific competition from conspecifics, other cerambycids, and other wood-inhabiting insects that can alter species distributions within trees [17]. In addition, most beetles are more likely to invade weakened pine trees; and the community structure of xylophagous beetles on pine trees was greatly changed after the colonization of pine wood nematode [18]. One of the most important factors causing a decline in pine tree vigor is the colonization of *B. xylophilus* and xylophagous insects [19–22]. Therefore, the diversity was also not only affected by the species of the host trees, but also the tree vigor. Study of biological ecological regulation of the xylophagous insects population plays an important role in successful implementation of chemical or biological control to vectors of PWN. However, the coexistence and competition relationship of these wood-inhabiting beetles was not clear enough in infected pines.

In this research, to understand the colonization sites (phloem and xylem) and colonization sequences of xylophagous beetles on pine trees, we investigated the species richness and spatial distribution of these beetles along PWN-infected tree boles during different stages of the disease. The analysis results of the spatial dynamic patterns of populations and the interspecific relationships between xylophagous beetles at different infection stages after PWN infection could provide valuable insights into the mechanisms of their change dynamics, which would help determine potential carriers of PWN. Understanding the interspecific relationships would also provide a basis for controlling the population of xylophagous beetles [23,24].

## 2. Materials and Methods

### 2.1. Study Area

From July to November 2019, we randomly selected pines infected with PWN in a *P. massoniana* forest of Fujian province and a *P. thunbergii* forest of Shandong Province. The infection stages were identified according to the growth of the pine trees infected by PWN, the colors of the pine needle leaves, the number of bite marks and wormholes, and the isolation of *B. xylophilus* [25–27]. We divided the *P. massoniana* and *P. thunbergii* infected by *B. xylophilus* into three stages: the initial stage of infection (weak wood); the medium stage of infection (dying wood); and the terminal stage of infection (a withered tree). In the initial stage of infection, some pine needles were yellow and withered, and the bole had insect bite marks, but the tree still had vitality and water, along with greater pine-resin exudation. In the medium stage of infection, the pine needles turned red and withered over a large area, the tree basically withered, the bole was dry, the borers increased, and the exudation of pine resin decreased. At the terminal stage, the needles largely fell off,

the bark was dry, there were pits and insect excreta in the phloem, the tree lost its water completely, and no resin exudation was observed (Figure A1).

### 2.2. Sampling Design

We cut trees close to the ground without damaging the tree crowns, and the residual root pile did not exceed 5 cm. We then measured and recorded the tree height and DBH. Low-speed drills were used to collect wood at 1.5 m from the base of the bole and at the crown, and pine wood nematodes were collected from wood samples using the Baermann funnel method and identified by the PCR test methods described in Wei et al. and morphological characteristics to determine whether the pine was infected with pine wood nematode disease [28,29]. Primers for partial ribosomal-DNA-large-subunit D2/D3 (LSU D2/D3) were forward-primer D2a (5′-ACAAGTACCGTGAGGGAAAGT-3′) and reverse-primer D3b (5′-TGCGAAGGAACCAGCTACTA-3′) [29]. The identification of *B. xylophilus* was based on diagnostic morphological characteristics. The whole pine was divided into three equal positions according to the height: the upper part, the middle part, and the bottom part. The felled boles were divided by chain saw into one-meter sections. We recorded the number of cavities on the felled bole; then, all insect samples were manually obtained directly from galleries of the infested pine trees using fine forceps; placed into centrifugal tube and store in 96-well frozen storage box; and labeled with the host pine, infection stage, feeding site, and location of the log. Three infected pines were sampled for each stage, and three replicates were used at each sampling site. A total of 27 infected pine trees was collected per site. We then calculated the volume of the truncated wood pine and counted the insect population in each meter of wood. We recorded the species, number, and distribution of insects in the bole. The xylophagous beetles were stored at 4 °C until use.

### 2.3. Xylophagous Beetle Identification

Morphological identification of the collected xylophagous beetles was accomplished under a stereomicroscope based on external morphological characteristics. These samples were preliminarily identified to the genera and species levels [27]. The cytochrome oxidase subunit I (COI) gene was used for molecular identification of the xylophagous beetles, as described in previous studies [30,31]. We extracted total DNA from 3–5 of each beetle according to morphology and placed each head separately in a centrifuge tube. Then, we used the CTAB method to extract DNA. Next, the purity and concentration of DNA were determined and the DNA was stored at −25 °C. The primers were LepF (5′-ATTCAACCAATCATAAAGATATTGG-3′) and HCO2198 (5′-TAAACTTCAGGGTGAC CAAAAAATCA-3′). The PCR reactions were conducted in a final volume of 50 μL containing 25 μL of 2 × Taq PCR Master Mix, 2.5 μL of 0.5 μmol/L of each primer, 10 μL of nuclease-free water, and 10 μL of the DNA template [30]. The PCR products were electrophoretically detected on agarose gel with a concentration of 1%. The PCR products were sequenced at the Boshang sequencing company of Fuzhou, and the sequences were deposited in GenBank (MT811991-MT812011). The homologous sequences were screened and the phylogenetic tree was constructed using the MEGA version 5.05 (Mega Limited, Auckland, New Zealand, 2011). For molecular identification, we used NCBI database to perform blast alignment on the obtained sequences, and the morphological identification was based on the described in "Insect Taxonomy" [32]. The PCR test methods described above was applied in the detection of pine wood nematode in xylophagous beetles collected in this research; thirty insects were detected in each species. All samples were detected for those insects with a low number not reaching thirty [30].

### 2.4. Functional Diversity

Functional diversity was calculated based on the locations and diets of the beetles using Rao's diversity index (Rao, 1982; Ricotta, 2005). To account for differences in the number of individuals collected at each site, species richness was rarefied using the Ve-

gan package, version 1.15-0, implemented in the R-statistical environment, version 2.7.1 (Lucent Technologies, Murray Hill, NJ, USA, 2008).

*2.5. Statistical Analyses*

To compare the species and abundance of xylophagous beetles under different forest types, the effects of different infected *P. massoniana* and *P. thunbergii* trees were analyzed through a one-way ANOVA test and an independent-sample T-test. The number of species (richness) and individual number (abundance) of xylophagous beetles on *P. massoniana* and *P. thunbergii* trees in different infection stages were calculated according to the positions of bole colonization by xylophagous beetles, and the Shannon diversity indices were calculated for the different infection stages. The calculation formulas were as follows:

$$\text{Niche breadth} : B_i = 1/\left(n\sum P_{ik}{}^2\right) a = 1, \tag{1}$$

where $B_i$ is the niche width of the $i$th species, $n$ is the value of the niche resource levels, and $P_{ik}$ is the proportion of the individuals of the $i$th species among the total number of individuals in the total resources.

$$\text{Niche overlap} : Q_{ij} = \sum (P_{ik}P_{jk})/\sqrt{\left(\sum P_{ik}{}^2 \sum P_{jk}{}^2\right)} \tag{2}$$

where $Q_{ij}$ is the niche overlap values of species $i$ and species $j$, and $P_{ik}$ and $P_{jk}$ are the proportion of individuals using the $k$th resource unit of species $i$ and species $j$, respectively, compared to the total number of corresponding species in the total resources.

We adopted the dominance index of Berger–Parker (1970):

$$d = N/N_t \tag{3}$$

where $d$ is the dominance index, $N$ is the number of individuals in a group, and $Nt$ is the number of all species.

$$H' = -\sum p_i \log p_i \tag{4}$$

where $H'$ is community diversity, and $p_i$ is the probability of the occurrence of species $i$. Community evenness was then calculated and analyzed using the formula proposed by Pielou (1966):

$$E = H'/\ln S \tag{5}$$

where $E$ is community evenness, $H'$ is community diversity, and $S$ is species richness. In the case of insect communities, species richness $S$ is the number of species in the community.

## 3. Results

*3.1. Identification, Composition, and Diversity of Xylophagous Insects in the Forests of P. thunbergii and P. massoniana*

The results of investigating *P. massoniana* and *P. thunbergii* infected with PWN were as follows: the average DBH of *P. massoniana* was 17.3 cm (14 m higher on average); the average DBH of *P. thunbergii* was 12 cm (12.3 m higher on average) (Table A1).

According to molecular identification of the COI gene and comprehensive identification of the morphological characteristics, we identified 20 species of insects. In this experiment (Figure 1), a total of 12,272 xylophagous beetles from 16 species with high richness were selected for analysis. Among them, nine species were taken from the *P. massoniana* forest and nine species were taken from the *P. thunbergii* forest. These species included six species in the family Curculionidae, subfamily Scolytinae, Ipidae, including *Xyleborus hawaiiensis* (Perkins, 1900), *Orthotomicus laricis* (Fabricius, 1792), *Orthotomicus erosus* (Nordl, 1888), *Pityogenes trepanatus* (Nördlinger, 1848), *Pityokteines spinidens* (Reitt, 1894), and *Cyrtogenius luteus* (Blandford, 1894), accounting for 53.70%; four species in the family Cerambycidae including *M. alternatus*, *Arhopalus rusticus* (Linnaeus, 1758), *Arhopalus*

*unicolor* (Gahan, 1906), and *Cephalallus oberthuri* (Sharp, 1905), accounting for 16.30%; three species in the family Elateridea including *Homotechnes brunneofuscus* (Nakane, 1954), *Dalopius vagus* (Brown, 1934), and *Tetrigus lewisi* (Candèze, 1873), accounting for 4.19%; two species in the family Curculionidae including *Shirahoshizo patruelis* (Voss, 1937) and *Hyposipalus gigas* (Fabricius, 1775), accounting for 18.26%; and one species in the family Cleridae (*Trichodes leucopsideus* Olivier, 1795), accounting for 18.26%. Other species belonged to Carabidae (1), Scarabaeidae (1), and Buprestidae (1).

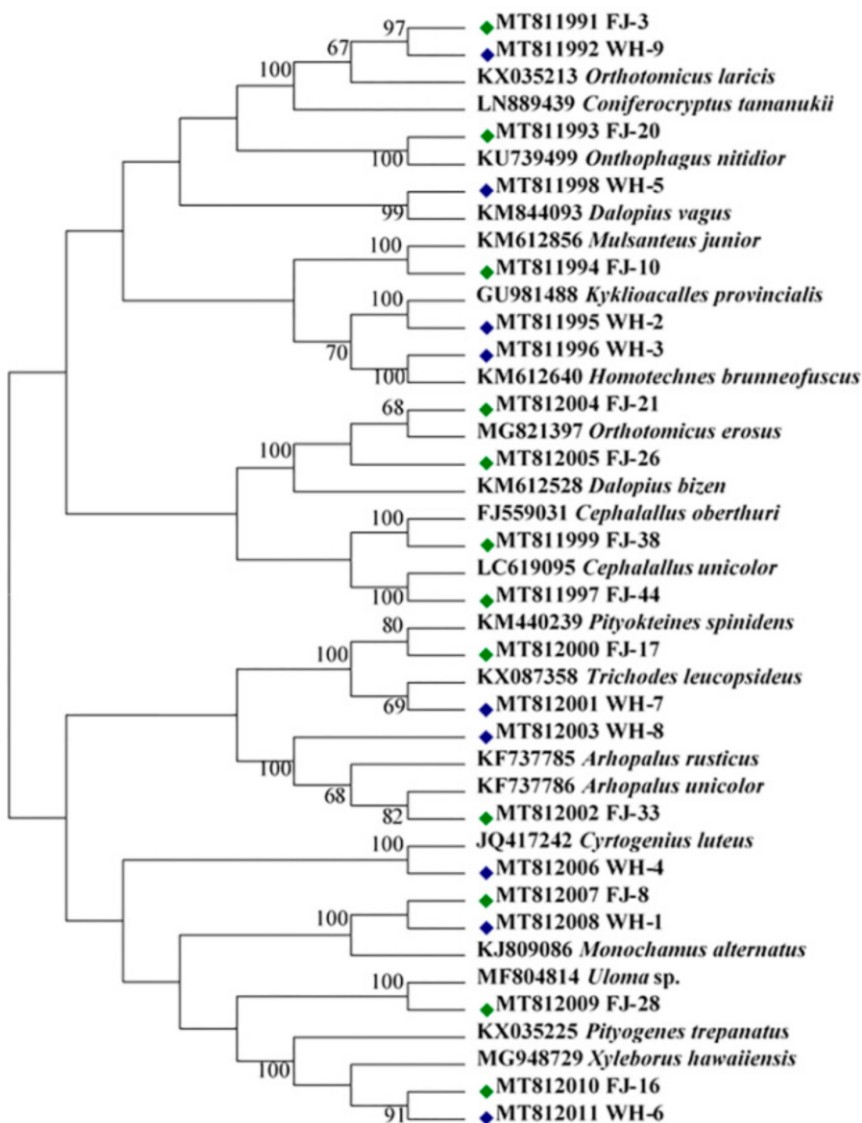

**Figure 1.** Phylogenetic tree based on the COI sequences of 20 xylophagous beetles collected from pine infected with PWN. FJ1–FJ12 labeled on green dots indicates xylophagous beetles collected from *P. massoniana* Lamb.; WH1–WH9 labeled on blue dots indicates xylophagous beetles collected from *P. thunbergii* Parl.; others indicate the reference sequences.

### 3.2. Rarefaction Curves Presenting the Relationship between the Number of Samples and Insect Species Richness in P. massoniana and P. thunbergii Forests

In this study, we analyzed a dilution curve constructed by the Shannon index (Figure 2). The results showed that the Shannon index at the species level increased with an increase in the amount of sampled data. When the amount of sampled data reached a certain value, the curve increased until becoming smooth. The sampling depth was sufficient, the amount of sampled data was reasonable, and most insect samples were covered.

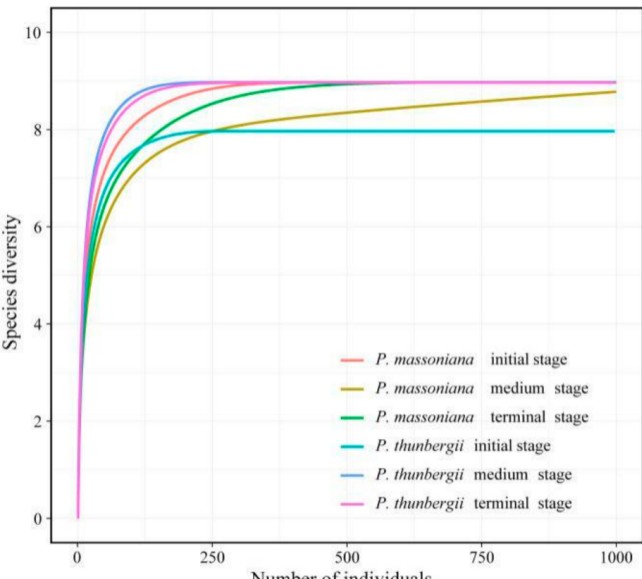

**Figure 2.** Rarefaction curves of the species richness level of *P. massoniana* and *P. thunbergii* at different affected stages. The red, yellow, and green curves represent the *P. massoniana* samples in three stages; the cyan, blue, and purple curves represent the *P. thunbergii* samples in three stages. Broken lines represent the 95% confidence interval for each curve.

### 3.3. Community Structure of Xylophagous Beetles at Different Infection Stages in the Forests of P. massoniana and P. thunbergii

In the *P. massoniana* forest, 11 insect species were identified: three species in the family Scolytinae, accounting for 54.52%; one species in the family Curculionidae, accounting for 24.29%; three species in the family Cerambycidae, accounting for 15.09%; one species in the family Cleridae, accounting for 3.15%; one species in the family Elateroidea, accounting for 0.93%; and other insects accounting for 2.01% (Figure 3).

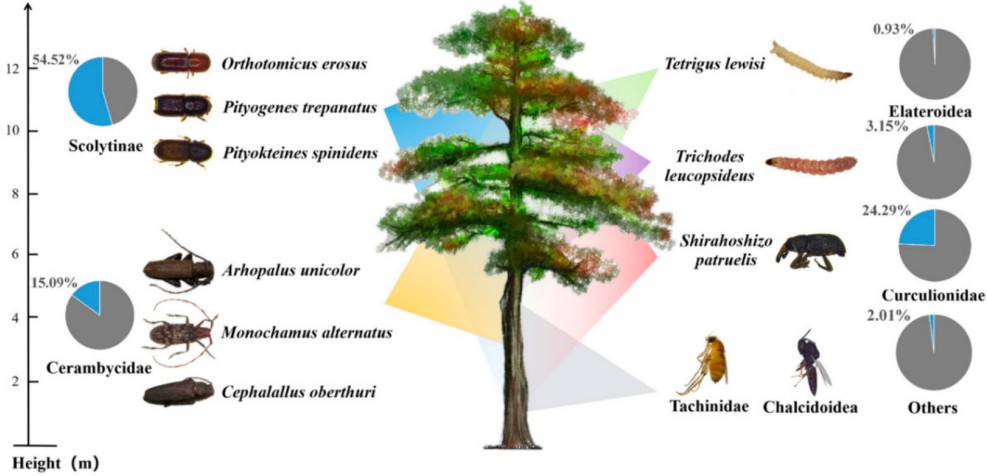

**Figure 3.** Diversity and distribution of xylophagous beetles and other insects in infected *P. massoniana*. The pie chart shows the proportion of xylophagous beetles, and the projection shows the distribution of xylophagous beetles.

In the *P. thunbergii* forest, nine insect species were identified: three species in the family Scolytinae, accounting for 51.43%; two species in the family Cerambycidae, accounting for 19.75%; one species in the family Cleridae, accounting for 14.14%; two species in the family Elateroidea, accounting for 13.33%; and one species in the family Curculionidae, accounting for 1.36% (Figure 4).

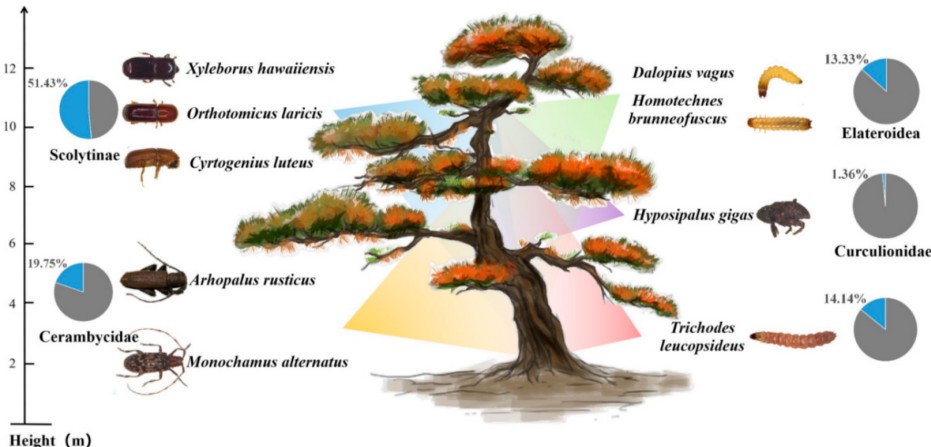

**Figure 4.** Diversity and distribution of xylophagous beetles in infected *P. thunbergii*. The pie chart shows the proportion of xylophagous beetles, and the projection shows the distribution of xylophagous beetles.

In the *P. massoniana* forest, *S. patruelis* and *O. erosus*, the two largest populations, were distributed in different stages of pine infection, while *M. alternatus*, an important vector of pine wood nematode disease, was enriched in the upper part of the initial stage of pine infection and the middle and bottom parts of the terminal stage of infection. In the *P. thunbergii* forest, *M. alternatus* was clustered in the bottom part of the bole during the medium and terminal stages of the disease (Figure 5).

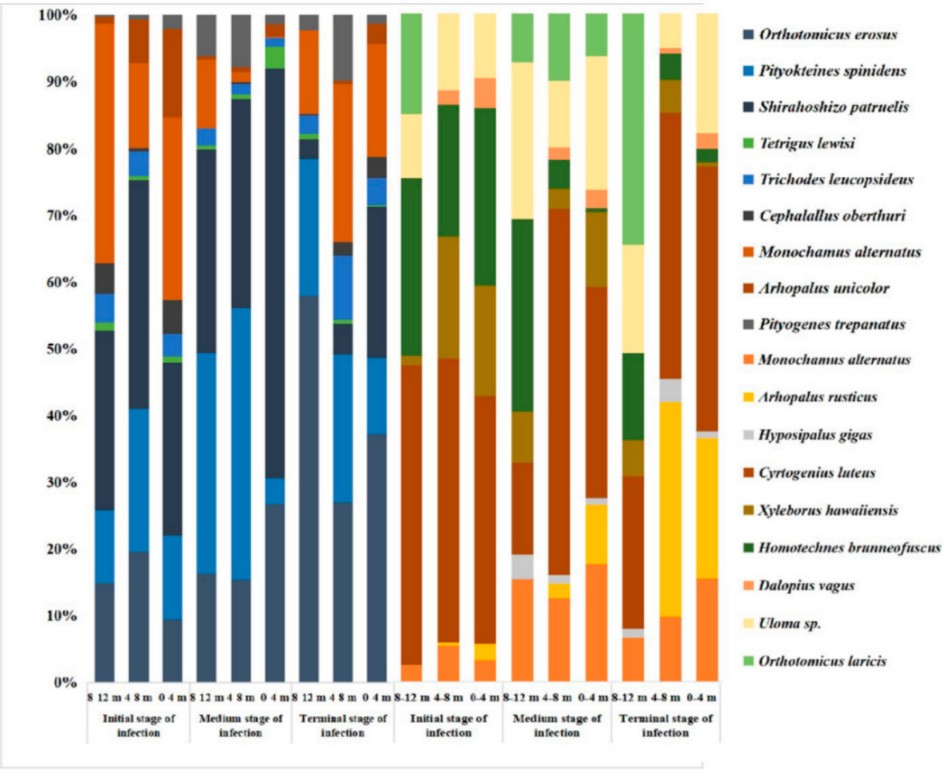

**Figure 5.** Composition of xylophagous beetles in the upper, middle, and bottom positions of *P. thunbergii* and *P. massoniana* infected by PWN at different stages.

### 3.4. Identification of Dominant Insect Species and Division of the Functional Groups

The dominant insect species of *P. massoniana* and *P. thunbergii* in different infection stages were analyzed according to the dominance index. The species were divided into

three different functional groups: woodboring beetles, bark beetles, and predators. In the *P. massoniana* and *P. thunbergii* forests infected with PWN, bark beetles dominated, with a dominance diversity of 0.8043 and 0.6611, respectively. *Orthotomicus* sp. was the dominant species, with a dominance index (*d*) of 0.3544 and 0.1261 in *P. massoniana* and *P. thunbergii*, respectively. Woodboring beetles mainly lived in weak trees infected with pine wood nematode, and their dominance indexes were 0.1510 and 0.1975, respectively. In this functional group, *M. alternatus* presented the largest dominance diversity of 0.8297 and 0.5416 in *P. massoniana* and *P. thunbergii*, respectively. The dominance of predators was 0.0617 and 0.1414, respectively, in the two forests (Tables 1 and 2).

The composition of xylophagous beetles in the two forests changed significantly in the different infection stages. The diversity of xylophagous beetles in the forest increased gradually with an increase of the infection stage in the *P. massoniana* forest (from 0.69 to 0.79, Table 3). According to the Shannon index, the diversity of xylophagous beetles was reduced in the terminal stage of infection in the *P. thunbergii* forest (0.66) and continued up to the terminal stage. In the medium stage, the diversity of xylophagous beetles in the *P. thunbergii* forest (0.76) was similar to that in the *P. massoniana* forest (0.79) (Figures A2 and A3 and Table 3).

**Table 1.** Function groups of insects and their dominance index (*d*) in *P. thunbergii* Parl.

| Function Groups | *d* (Functional Group) | Dominant Insect Species | *d* (Insects) |
|---|---|---|---|
| woodboring beetles | 0.1975 | *Monochamus alternatus* | 0.5416 |
| | | *Arhopalus rusticus* | 0.4584 |
| bark beetles | 0.6611 | *Hyposipalus gigas* | 0.0206 |
| | | *Cyrtogenius luteus* | 0.5570 |
| | | *Xyleborus hawaiiensis* | 0.0947 |
| | | *Homotechnes brunneofuscus* | 0.1777 |
| | | *Dalopius vagus* | 0.0239 |
| | | *Orthotomicus laricis* | 0.1261 |
| predators | 0.1414 | *Trichodes leucopsideus* | 1.0000 |

**Table 2.** Function groups of insects and their dominance index (*d*) in *P. massoniana* Lamb.

| Function Groups | *d* (Functional Group) | Dominant Insect Species | *d* (Insects) |
|---|---|---|---|
| woodboring beetles | 0.1510 | *Monochamus alternatus* | 0.8297 |
| | | *Arhopalus unicolor* | 0.1055 |
| | | *Cephalallus oberthuri* | 0.0648 |
| bark beetles | 0.8043 | *Orthotomicus erosus* | 0.3544 |
| | | *Pityokteines spinidens* | 0.2863 |
| | | *Shirahoshizo patruelis* | 0.3082 |
| | | *Pityogenes trepanatus* | 0.0511 |
| | | *Monochamus alternatus* | 0.8297 |
| predators | 0.0617 | *Tetrigus lewisi* | 0.1536 |
| | | *Trichodes leucopsideus* | 0.5210 |

**Table 3.** Community characteristic indexes of xylophagous beetles in *P. massoniana* and *P. thunbergii* at different infection stages.

| Infected Stages | *P. massoniana* | | | *P. thunbergii* | | |
|---|---|---|---|---|---|---|
| | *H′* | *E* | *S* | *H′* | *E* | *S* |
| Initial stage | 0.69 ± 0.06 | 0.35 ± 0.02 | 7 | 0.79 ± 0.07 | 0.36 ± 0.04 | 9 |
| Medium stage | 0.79 ± 0.03 | 0.38 ± 0.02 | 8 | 0.76 ± 0.03 | 0.37 ± 0.06 | 8 |
| Terminal stage | 0.79 ± 0.02 | 0.38 ± 0.01 | 8 | 0.66 ± 0.06 | 0.30 ± 0.06 | 9 |

Note: Data are means ± standard error (SE), *n* = 3.

### 3.5. Interspecific Competition among Xylophagous Beetles at Different Infection Stages in P. thunbergii and P. massoniana Forests

By comparing the niche breadth of the nine species of xylophagous beetles with the greatest population richness in each forest, we found that *D. vagus* occupied the main niche in the *P. thunbergii* forest (0.993). In the *P. massoniana* forest, *M. alternatus* had the largest niche breadth (0.909) (Table A2). In the *P. thunbergii* forest, *T. leucopsideus* and *C. luteus* firmly occupied the middle and bottom positions of the bole in the medium and terminal stages of the infected pine trees, spatially following *M. alternatus*, while *O. laricis* (Q = 0.971) competed with *M. alternatus* in the middle part of the bole at the specific medium stage of the disease and showed obvious aggregation behavior in the upper part of the bole at the initial and terminal stages of the disease. The niche overlap index of *O. laricis* and *M. alternatus* gradually increased over the duration of infection. Niche overlap also existed among *P. spinidens*, *O. erosus*, and *T. lewisi* (Q > 0.969) (Figure 6, Table A3). In addition, *O. laricis* and *A. rusticus* showed the largest niche overlap index, which mainly occurred in the middle and bottom parts of the bole in the terminal infection stage.

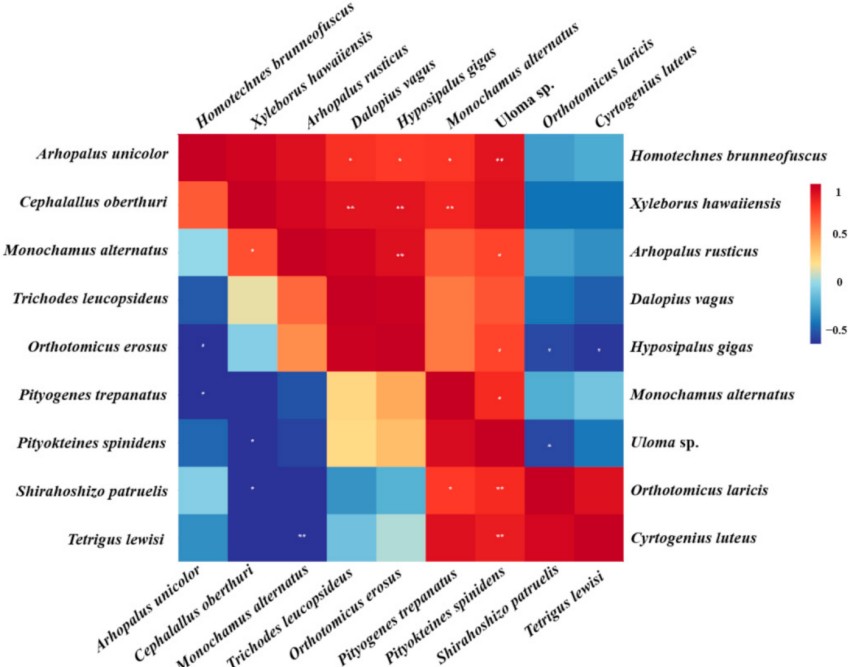

**Figure 6.** Correlation heat map analysis of xylophagous beetles in *P. massoniana* and *P. thunbergii*. The red color of the grid in the figure indicates that the niche overlap index between vertical and horizontal species is larger, and the blue color indicates that the niche overlap index between species is smaller; and * indicates significant ** indicates extremely significant.

### 3.6. Identification of PWN in Xylophagous Beetles

According to the result of the PCR test and morphological characteristics, *B. xylophilus* was isolated from *M. alternatus* in both *P. massoniana* and *P. thunbergia*. No *B. xylophilus* was isolated from other xylophagous beetles and other insects in this research.

## 4. Discussion

In pine forests infected by *B. xylophilus*, the complex invasions of xylophagous beetles accelerate the death of pine trees, reduce the resistance of pine trees to *B. xylophilus*, and provide intermediate hosts for the transmission of PWN [33]. In our study, the composition and distribution of xylophagous beetles were surveyed in *P. massoniana* and *P. thunbergia* trees in different infection stages. Nine species of xylophagous beetles were identified in *P. massoniana* and *P. thunbergii*, respectively. In addition, two parasites were identified in *P. massoniana*. These parasites shared two of the same taxa: *M. alternatus* and *T. leucopsideus*

*M. alternatus* and Cleridae were also reported in *Pinus yunnanens* infected by PWN [34]. Moreover, in the present study, no *B. xylophilus* was isolated from these xylophagous beetles except for *M. alternatus*. However, *O. erosus* and *Pityogenes* sp. were reported as carriers of *Bursaphelenchus* spp. [35]. More studies are needed to determine if there are other carriers of *B. xylophilus* besides *Monochamus*.

The composition and structure of xylophagous beetles were different between the two host trees species in different infection stages. The diversity of insects was affected by multiple biotic and abiotic factors, such as climate, topography, soil, host plants, and natural enemies [36]. Together, these complex factors influenced the distribution and population of xylophagous beetles in the two forests. Pine trees infected by *B. xylophilus* have different weakening cycles when observed in different forests, different stand ages, and different management modes, each showing different attraction effects to xylophagous beetles. For example, Blatt et al. [37] caught greater numbers of several *Monochamus* species in Christmas tree plantations than in adjacent forests.

Interestingly, the population of *M. alternatus*, an important vector of pine wood nematode disease, was enriched in the upper part of *P. massoniana* during the initial infection stage, while the middle and bottom parts were enriched in the terminal infection stage. In *P. thunbergii*, *M. alternatus* also tended to clustered in the bottom part during the medium and terminal infection stages. This phenomenon might be related to the host tree's utilization of nutritional resources via vertical migration [22]. *T. leucopsideus* and *C. luteus* were also enriched in the middle and bottom sections of the bole in the medium and terminal stages, possibly because the complex colonization of other xylophagous beetles increased the pressure of nutritional competition in the crown layer during the medium and terminal infection stages. Additionally, during the terminal stage of PWN infection, pine trees lost a great deal of water and nutrients. The canopy of the pine tree then became unsuitable for xylophagous beetles, so they migrated to the bottom part of the bole [38]. Akbulut et al. [17] also believed that the loss of water from pine bole might reduce the survival rate of xylophagous beetles.

The colonization order of insect functional groups living on pine trees also followed a certain law (from health to infection to death) for the pine wood nematode. The relative richness of *O. erosus* was found to be the highest in xylophagous beetles in *P. massoniana*. Moreover, *O. erosus* was reported to be a pioneer species [39] able to reduce the resistance and tree potential of the host pine. Subsequently, other xylophagous beetles and predators gradually enter the tree. In the different stages of infection with pine wood nematode disease (until death), insects with different functional groups occupy the dominant position. According to the dynamic analysis of the niche, *M. alternatus*, as a secondary pest, damaged the weakened pine trees after colonization with pioneer bark beetles such as *C. luteus* and *O. erosus.* Thus, colonization with *M. alternatus* might benefit from bark beetles. The presence of exit holes, galleries, and tunnels made by the primary colonizers subsequently has a positive effect on the species richness and abundance of secondary colonizers and their predators [40,41]. It was previously reported that bark beetle reaches its niche load in terms of its trophic niche during the medium stage of infection, which limits increases in the *M. alternatus* population [42]. Therefore, the colonization of other xylophagous beetles might affect the distribution of the *M. alternatus* population.

In the initial stage of infection, *M. alternatus* was found to be mainly in competition with Scolytidae, which was able to overcome the resistance of the host pine tree and further reduce the tree's vigor. In the terminal stage of infection, competition with *M. alternatus* was mainly provided by species such as Elateridae. At this stage, xylophagous beetles in the bole mainly played a role in accelerating the decline of the pine trees [43]. We suspect that intraspecific competition, cannibalism, and resource quality affected the survival of *M. alternatus*; otherwise, the species without a competitive advantage would be eliminated early [44].

The niche overlap between *O. laricis* and *M. alternatus* accompanied the entire infection cycle and peaked at the terminal stage. Scolytinae had a certain inhibitory effect on the

canopy of *M. alternatus*, which together limited the population growth of *M. alternatus*. Research on xylophagous beetles in *P. massoniana* in Zhejiang also found that *Tomicus piniperda* (Scolytinae) inhibited population growth of the *M. alternatus* [18]. The population dynamics of *M. alternatus* might be related to the resistance of bark beetles to host trees and competition with other xylophagous beetles. In the *P. massoniana* forest, *O. erosus* and *T. leucopsideus* showed an obvious spatial migration phenomenon, suggesting that *T. leucopsideus* might be a predator of *O. erosus*. Ultimately, understanding the spatial distribution and interspecific relationships of xylophagous beetles in pine trees infected with *B. xylophilus* provides a basis for regulating the population of PWN vectors to control the transmission of PWD.

## 5. Conclusions

The diversity of xylophagous beetles was different not only between the two host pine trees but also among the different infection stages. *M. alternatus* and *T. leucopsideus* were investigated in both pines. *M. alternatus*, as the only vector of *B. xylophilus* in our research, occupied a large niche and showed a similar population fluctuation during the three infection stages in both host trees. It might be related to the cooperation and competition of other xylophagous beetles to the host defense and nutritional utilization.

**Author Contributions:** Data curation, X.C. and Q.M.; Investigation, X.C.; Methodology, X.H.; Project administration, F.Z.; Resources, X.H.; Supervision, M.Y., G.L. (Guoqiang Li), J.L., G.L. (Guanghong Liang), S.W., R.W. and X.H.; Writing—original draft, X.C. and Q.M.; Writing—review & editing, Q.M. and X.H. All authors have read and agreed to the published version of the manuscript.

**Funding:** This research was funded by the National Key Research and Development Program of China (grant number 2017YFD0600105); the National Natural Science Foundation of China (grant numbers 31800548); the Forestry Science Research Project of Fujian Forestry Department (grant number Minlinke [2019] 16); and the Forest Science Peak Project of College of Forestry, Fujian Agriculture and Forestry University (grant numbers 71201800743).

**Institutional Review Board Statement:** Not applicable.

**Informed Consent Statement:** Written informed consent has been obtained from the patient(s) to publish this paper.

**Data Availability Statement:** The data presented in this study are openly available in Zenodo at 10.5281/zenodo.5235663.

**Conflicts of Interest:** The authors declare no conflict of interest.

## Appendix A

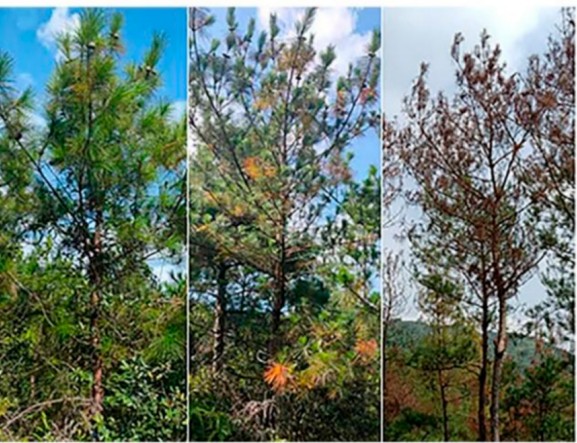

**Figure A1.** Three stages of pine infected with pine wood nematode disease. From left to right are the initial stage, medium stage, and terminal stage of the disease.

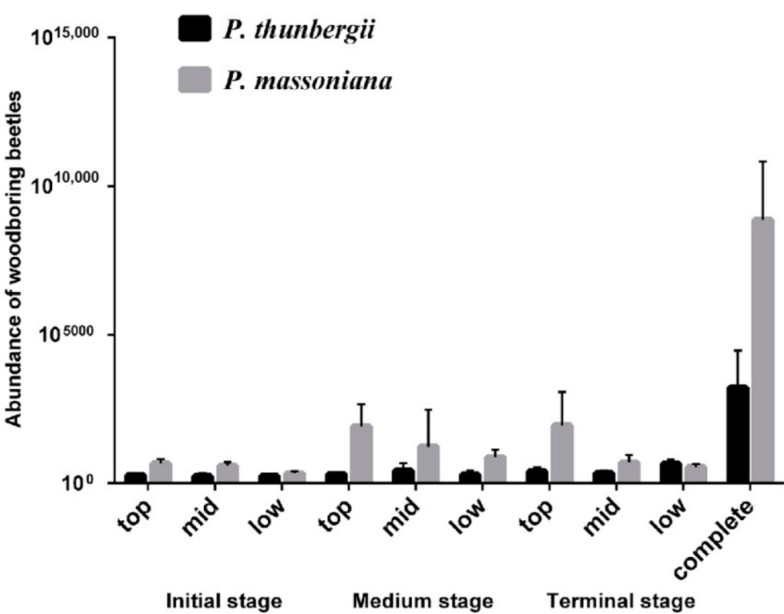

**Figure A2.** Abundance of xylophagous beetles in different infected stage.

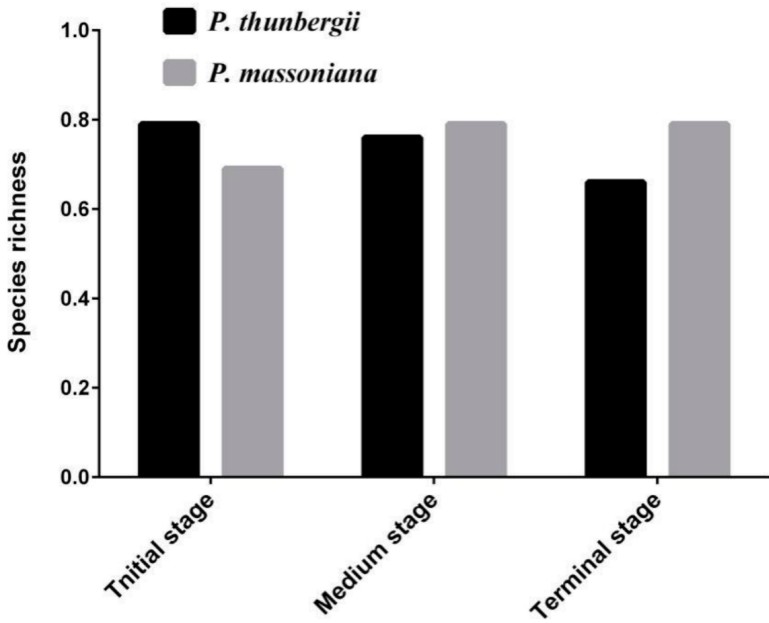

**Figure A3.** Species richness of xylophagous beetles in different infected stages.

**Table A1.** The average height, diameter at breast height (DBH), and annual ring of the sample pine trees in the infected *P. thunbergii* and *P. massoniana*.

| Host Tree | Forest Composition | Infected Stage | Average DBH (cm) | Average Hight (m) | Annual Ring (Year) |
|---|---|---|---|---|---|
| *Pinus thunbergii* | Pure forest | Initial stage | 9.26 ± 3.61 | 10.25 ± 2.523 | 18.7 ± 4.2 |
| | | Medium stage | 11.50 ± 1.53 | 12.01 ± 2.650 | 20.3 ± 2.5 |
| | | Terminal stage | 16.31 ± 2.65 | 15.24 ± 2.082 | 26.7 ± 4.0 |
| *Pinus massoniana* | Mingled forest | Initial stage | 15.44 ± 2.52 | 12.63 ± 2.659 | 15.0 ± 2.9 |
| | | Medium stage | 17.58 ± 3.51 | 13.52 ± 2.009 | 20.0 ± 1.1 |
| | | Terminal stage | 20.19 ± 4.51 | 17.17 ± 2.520 | 20.1 ± 3.6 |

**Table A2.** Niche breadth of xylophagous beetles in the infected *P. thunbergii* and *P. massoniana*.

| Beetles in *P. thunbergii* | Niche Breadth | Beetles in *P. massoniana* | Niche Breadth |
|---|---|---|---|
| *Monochamus alternatus* | 0.777 | *Monochamus alternatus* | 0.909 |
| *Cyrtogenius luteus* | 0.967 | *Orthotomicus erosus* | 0.713 |
| *Homotechnes brunneofuscus* | 0.879 | *Pityokteines spinidens* | 0.761 |
| *Orthotomicus laricis* | 0.831 | *Shirahoshizo patruelis* | 0.630 |
| *Arhopalus rusticus* | 0.446 | *Pityogenes trepanatus* | 0.653 |
| *Hyposipalus gigas* | 0.661 | *Tetrigus lewisi* | 0.729 |
| *Xyleborus hawaiiensis* | 0.927 | *Trichodes leucopsideus* | 0.871 |
| *Dalopius vagus* | 0.993 | *Cephalallus oberthuri* | 0.703 |
| *Trichodes leucopsideus* | 0.892 | *Arhopalus unicolor* | 0.716 |

**Table A3.** Niche overlap of xylophagous beetles in the infected pine trees.

| Beetles in *P. thunbergii* | Label | Beetles in *P. massoniana* | Label |
|---|---|---|---|
| *Monochamus alternatus* | 1 | *Monochamus alternatus* | A |
| *Cyrtogenius luteus* | 2 | *Orthotomicus erosus* | B |
| *Homotechnes brunneofuscus* | 3 | *Pityokteines spinidens* | C |
| *Orthotomicus laricis* | 4 | *Shirahoshizo patruelis* | D |
| *Arhopalus rusticus* | 5 | *Orthotomicus erosus* | E |
| *Hyposipalus gigas* | 6 | *Tetrigus lewisi* | F |
| *Xyleborus hawaiiensis* | 7 | *Trichodes leucopsideus* | G |
| *Dalopius vagus* | 8 | *Cephalallus oberthuri* | H |
| *Trichodes leucopsideus* | 9 | *Arhopalus unicolor* | I |

| Niche overlap index | | | |
|---|---|---|---|
| Q12 | 0.933 | Qab | 0.881 |
| Q13 | 0.681 | Qac | 0.726 |
| Q14 | 0.971 | Qad | 0.576 |
| Q15 | 0.812 | Qae | 0.661 |
| Q16 | 0.990 | Qaf | 0.704 |
| Q17 | 0.741 | Qag | 0.960 |
| Q18 | 0.900 | Qah | 0.925 |
| Q19 | 0.986 | Qai | 0.804 |
| Q23 | 0.878 | Qbc | 0.811 |
| Q24 | 0.970 | Qbd | 0.566 |
| Q25 | 0.788 | Qbe | 0.807 |
| Q26 | 0.521 | Qbf | 0.736 |
| Q27 | 0.899 | Qbg | 0.976 |
| Q28 | 0.994 | Qbh | 0.647 |
| Q29 | 0.980 | Qbi | 0.460 |
| Q34 | 0.735 | Qcd | 0.936 |
| Q35 | 0.446 | Qce | 0.992 |
| Q36 | 0.578 | Qcf | 0.990 |
| Q37 | 0.991 | Qcg | 0.826 |
| Q38 | 0.923 | Qch | 0.465 |
| Q39 | 0.785 | Qci | 0.559 |
| Q45 | 0.945 | Qde | 0.910 |
| Q46 | 0.959 | Qdf | 0.975 |
| Q47 | 0.767 | Qdg | 0.628 |
| Q48 | 0.939 | Qdh | 0.381 |
| Q49 | 0.986 | Qdi | 0.624 |
| Q56 | 0.849 | Qef | 0.969 |
| Q57 | 0.452 | Qeg | 0.795 |
| Q58 | 0.628 | Qeh | 0.366 |
| Q59 | 0.812 | Qei | 0.446 |
| Q67 | 0.643 | Qfg | 0.777 |
| Q68 | 0.841 | Qfh | 0.475 |
| Q69 | 0.959 | Qfi | 0.624 |

**Table A3.** *Cont.*

| Beetles in *P. thunbergii* | Label | Beetles in *P. massoniana* | Label |
|---|---|---|---|
| Q78 | 0.939 | Qgh | 0.784 |
| Q79 | 0.829 | Qgi | 0.642 |
| Q89 | 0.960 | Qhi | 0.994 |

Note: 1–9 represents nine species of xylophagous beetles in the *P. thunbergii* forest, A–I represents nine species of xylophagous beetles in the *P. massoniana* forest.

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
