# Peer review of "Diversity and Distribution of Xylophagous Beetles from Pinus thunbergii Parl. and Pinus massoniana Lamb. Infected by Pine Wood Nematode"

_forests, doi:10.3390/f12111549_

Round 1

Reviewer 1 Report

Review of the article titled “Diversity and distribution of xylophagous beetles from Pinus thunbergii and Pinus massoniana infected by pine wood nematode” (ID: forests-1432734).  Chu et al. present an interesting survey on dendrophilic beetle communities from two pine host trees distributed in northern and southern China. This is not simply a faunistic study, as it examines the variability of xylophagous beetle species in respect to various factors, incl. tree host and vertical layers of the tree. The concept of the study is to discover the beetle diversity of pines infected by pinewood nematode Bursaphelenchus xylophilus being a pest of pines and having an important impact on timber harvesting (Pine wilt disease). This study could have a great impact on the understanding of the transmission of nematodes by xylophagous beetles, but unfortunately, this obvious problem is not included in the paper (see comments below). I appreciate the effort made in studying beetle communities along tree trunks, which is challenging. Generally, I find this study as potentially publishable in Forests, however, there are several less or more serious problems, which need to be solved by Authors during the revision of their work.

First of all, the whole Introduction is focused on PWN/PWD, whereas the study is not on this nematode or disease, and is only superficially related to it. PWN/PWD is a nice acknowledgement for this study, however, the accent of the study should be changed or additional data and analyses should be added.

  1. 70-71

I dislike such statements suggesting that xylophagous beetles are destructive agents in forests. Dendrophilic beetles are natural elements of forest ecosystems and play their crucial roles. Only in some conditions, particularly in tree plantations, some of these beetles are detrimental to the wood of planted trees, but not for the forest as the habitat.

2.1.

Study area Major drawback of the study design is that it did not include nematode-free forest stands for comparison (and control). Understanding xylophagous beetle communities' diversity is not possible without knowing the species richness and abundance in natural and healthy forests.

2.2. Sampling design

PWN was collected from pine trees but there is no information about analyses in which that information was used.

Especially I wonder why the PCR test was not used to verify the presence of PWN in collected beetles? This would be a really interesting and important finding to determine beetle hosts, which transmit this nematode. Authors already have DNA isolates from beetles (for barcoding), so it should be possible to amplify some markers for the nematode and extend this study.

  1. 116-117

I do not understand this sentence. What does it mean that 3-4 species were morphologically determined? And why were these beetles preserved jointly in a single tube?

  1. 126-127

Molecular methods used for species barcoding are very superficial and have to be better explained. What kind of phylogenetic tree, constructed with the use of which method? Why species delimitation methods were not implemented for the assignment of individuals to species?

Was any reference used for species identification? Some general databases (NCBI, BOLD?) or a database for beetles from Chinese pines was first developed?

  1. 167 and l. 274 - 9 species found in P. massoniana vs l. 197 - 11 species found in P. massoniana – so how many species from this host tree?

Moreover, fig.3 and fig. 4 suggest different numbers of species as some were identified not to species level but to family or genus level, so it should be reported number of taxa (not species).

  1. 169

Scolytinae is a subfamily and should be listed along with other Curculionidae if this part describe species richness in families of beetles

  1. 172

What is  A. rusticus? – no genus name is specified

Fig. 1 is very unclear. I suggest showing a standard phylogenetic tree and rooting it in cox1 from some insect-related beetles. Moreover, the presentation of just accession numbers from GenBank does not say anything. Why are some of the sequenced beetles also not species determined?

Fig. 5 is a nice picture, however, it is hardly readable as some colors seemed to be repeated for other taxa (e.g. orange).

Fig. 6

Something is wrong with this graph. E.g. Hyposipalus and Monochamus are in reverse order on both axes.

What are white dots in some squares?

Tab. 1, A2, A3 Scolytoplatypus sp. had to be in italic.

Tab. 2 What about Parasitic wasp and Tachinidae – why are these not-beetle taxa reported there?

  1. 277-279

Maybe I missed this information but where are descriptions of PWN identification methods and results in examined beetles? It is a very important finding but how is it supported by the study? Maybe it is finding not from this study but literature (see l. 280)?

Author Response

Reviewer 1

General comment 1: 

First of all, the whole Introduction is focused on PWN/PWD, whereas the study is not on this nematode or disease, and is only superficially related to it. PWN/PWD is a nice acknowledgment for this study, however, the accent of the study should be changed or additional data and analyses should be added.

Response: We appreciate very much your comments. We restated the introduction to focus on xylophagous beetles rather than PWN and PWD.

SPECIFIC COMMENTS:

Comment 1: Line 70-71: I dislike such statements suggesting that xylophagous beetles are destructive agents in forests. Dendrophilic beetles are natural elements of forest ecosystems and play their crucial roles. Only in some conditions, particularly in tree plantations, some of these beetles are detrimental to the wood of planted trees, but not for the forest as the habitat.
Response: It is undeniable that wood beetles play a very important and irreplaceable role in the overall forest ecosystem. But we think that during the occurrence of PWD, the outbreak of PWN population and the colonization of xylophagous beetles were mutually promoted, and the tree weakness was caused by both. Therefore we modified the content to “In addtion, most of beetles are more likely to invade weakened pine trees. One of the most important factors causing a decline in pine tree vigor is the colonization of B. xylophilus and xylophagous insects”(line 67-69 in the revised version).

Comment 2: 2.1: Study area Major drawback of the study design is that it did not include nematode-free forest stands for comparison (and control). Understanding xylophagous beetle communities' diversity is not possible without knowing the species richness and abundance in natural and healthy forests.

Response:  The main objective of this experiment was to investigate the species and distribution of xylophagous beetles in pine trees at different stages of disease but not a whole forest system. The species and distribution of xylophagous beetles on P. massoniana forests uninfected with pine wood nematode disease have been studied previously. Therefore, we did not set up a control group of uninfected pines. Potential host insects of pine wood nematodes are expected to be identified in terms of temporal dynamics. Therefore, pine trees that are not infected with pine wood nematode disease are not the focus of this experiment.

Comment 3: 2.2: Sampling design

PWN was collected from pine trees but there is no information about analyses in which that information was used.

Especially I wonder why the PCR test was not used to verify the presence of PWN in collected beetles? This would be a really interesting and important finding to determine beetle hosts, which transmit this nematode. Authors already have DNA isolates from beetles (for barcoding), so it should be possible to amplify some markers for the nematode and extend this study.
Response: We have identified PWN from the trunk to demonstrate that the pine trees was infected with pinewood nematode disease by Berman funnel method. We have isolated nematodes from the adults of xylophagous beetles using the the PCR test,  but no B. xylophilus was isolated from these xylophagous beetles except for M. alternatus. So we just mentioned this result in discussion part. We added a sentence in 2.2 and 2.3 in MM to describe the identification of PWN, a sentence in result part to state the result of identification of PWN form xylophagous beetles in the revised version.

Comment 4: Line 116-117: I do not understand this sentence. What does it mean that 3-4 species were morphologically determined? And why were these beetles preserved jointly in a single tube?
Response: We have revised this vague expression. This sentence was changed to “We extracted total DNA from 3-5 of each beetle according to morphology and placed each head separately in a centrifuge tube”(line 127-129 in the revised version).

Comment 5: Line 126-127: Molecular methods used for species barcoding are very superficial and have to be better explained. What kind of phylogenetic tree, constructed with the use of which method? Why species delimitation methods were not implemented for the assignment of individuals to species?

Was any reference used for species identification? Some general databases (NCBI, BOLD?) or a database for beetles from Chinese pines was first developed?
Response: The phylogenetic tree was constructed by mega 6.0 software and maximum likelihood method (ML). The bootstrap test was repeated for 1000 times, and the other parameters were the default.

For molecular identification, we use NCBI database to perform blast alignment on the obtained sequences, and the morphological identification is based on “Insect Taxonomy” (Cai Banghua et al, 2017)

Comment 6: 167 and l. 274 - 9 species found in P. massoniana vs l. 197 - 11 species found in P. massoniana – so how many species from this host tree?

Moreover, fig.3 and fig. 4 suggest different numbers of species as some were identified not to species level but to family or genus level, so it should be reported number of taxa (not species).
Response: We analyzed 9 representative species with the largest populations in each of the two forests. Two of the species are duplicates, so the total number of species is 16. The P. massoniana forest also included two natural enemy insects, but they were not included in the data of subsequent analysis and were only presented as research findings. Line 219: The word “insect” is used instead of “xylophagous beetles”.

Comment 7: Line 169: Scolytinae is a subfamily and should be listed along with other Curculionidae if this part describe species richness in families of beetles.
Response: This part describes the species richness of beetle family. We change “Scolytinae” to “Scolytidae”(line 187 in the revised version). The names of all departments were revised uniformly.

Comment 8: Line 172: What is A. rusticus? – no genus name is specified.
Response: We supplemented the Latin name of A. rusticus, added the appraiser and year (line 191 in the revised version). And we proofread all species names, and supplemented the missing information.

Comment 9: Fig. 1 is very unclear. I suggest showing a standard phylogenetic tree and rooting it in cox1 from some insect-related beetles. Moreover, the presentation of just accession numbers from GenBank does not say anything. Why are some of the sequenced beetles also not species determined?
Response: We have modified figure 1. We identified all xylophagous beetles to species and reused adjacency methods to construct a standard phylogenetic tree, including beetle gene sequences from P. massoniana and P. thunbergii forests, as well as reference species from NCBI database. The approval rate of each branch was added, and the target gene number was marked in Figure Legends.

Comment 10: Fig. 5 is a nice picture, however, it is hardly readable as some colors seemed to be repeated for other taxa (e.g. orange).
Response: We changed the color of the bar chart and legend in Figure 5 to eliminate confusing expressions.

Comment 11: Fig. 6

Something is wrong with this graph. E.g. Hyposipalus and Monochamus are in reverse order on both axes.

What are white dots in some squares?
Response: We corrected the wrong order in Figure 6. The lower left part of the figure represents the correlation coefficient between 9 species of xylophagous beetles in the P. massoniana forest, and the upper right part represents the correlation coefficient between 9 species of xylophagous beetles in P. thunbergii forest. The white asterisk in the figure indicates the significance of the correlation, with * indicating significant and ** indicating extremely significant. We added a caption to clarify the meaning of the picture.

Comment 12: Tab. 1, A2, A3 Scolytoplatypus sp. had to be in italic.
Response: We corrected all the Latin words in the manuscript and italicized the names of the genera and species.

Comment 13: Tab. 2 What about Parasitic wasp and Tachinidae – why are these not-beetle taxa reported there?
Response: Parasitic wasp and Tachinidae are two natural enemy insects found in the P. massoniana forest, which do not belong to the beetle taxa. We have deleted them from Table 2, so that they also correspond to the 9 xylophagous beetles mentioned above.

Comment 14: Line 277-279: Maybe I missed this information but where are descriptions of PWN identification methods and results in examined beetles? It is a very important finding but how is it supported by the study? Maybe it is finding not from this study but literature (see l. 280)?
Response: Because most woodborers except Scolytidae are in the larval stage and cannot be raised in captivity to become adults, they cannot carry nematodes even if they are successfully raised in captivity. We isolated nematodes from the adults of xylophagous beetles using the Berman funnel method and the PCR test method, but unfortunately no B. xylophilus was isolated from these xylophagous beetles except for M. alternatus.

Reviewer 2 Report

The work is well written, nicely designed and properly evaluated. I especially evaluate PCR analysis supplementing morphological determinations. The work does not contain any hypothesis in the introduction, which is not to the detriment, because it is descriptive and provides important data. The range of studied samples, both trees and insects, is sufficient and above standard for this type of work. The graphical representation of the results is perfect. I have only a few formal remarks and recommendations at the end.

Lines 83 -85 We divided the P. massoniana and P. thunbergii infected 83 by B. xylophilus into three stages: the initial stage of infection (weak wood); the medium 84 stage of infection (dying wood); and the terminal stage of infection (a withered tree)….Please add pictures of stages

Line 94 We cut trees close to the ground without damaging the tree crowns…How did you protect crown?

Lines 161-163 Move these lines into methods.

Move the Fig 3 and Fig 4 labels to the previous pages.

Line 231 M. alternatus has a larger font.

Lines 243 Move on next page.

Table 1 Scolytoplatypus has a larger font, change on italics.

Table 2 Clearidae sp. and Tachinidae. Unite, either with „sp.“ or without (similarly line 253, 276, 277). If you want to use „sp.“ then use „spp.“ There were certainly more species or was it one species? Why was it not determined? Perhaps the use of the English „Clerid“ is also worth considering.

Line 290 Move [34] behind „Blatt et al“. Is dot missing? The same on line 303.

Please provide a summary of your study at the end of the discussion.

Round 2

Reviewer 1 Report

Authors of the manuscript entitled “Diversity and distribution of xylophagous beetles from Pinus thunbergii and Pinus massoniana infected by pine wood nematode” (ID: forests-1432734) responded to my previous comments. I appreciate their effort in revising their work. I find most of the responses convincing. However, I still think that this kind of study should contain analyses of Bursaphelenchus uninfected tree hosts, as without such data it is not possible to understand how infection affect dendrophilic beetle communities, especially as according to this study, no beetle species (except single longhorn beetle) was vector of this nematode. So, it is unclear if beetle communities change because of the presence of nematodes having impact on the host plant (of both – nematodes and beetles) or if there are some other factors shaping beetle communities. Moreover, I disagree that this study examines temporal changes of beetle communities. To do this, the same host tree should be investigated from health until death, which is basically impossible. This study compares beetle communities from trees of different stages caused by nematode infection. I suggest being specific and avoiding the term “temporal”.

Minor comment. As far as I know, in the current taxonomy of beetles Scolytinae is a subfamily, not family, so you should not simply change the scientific name of this taxon.

Author Response

Reviewer 1

General comment 1

Authors of the manuscript entitled “Diversity and distribution of xylophagous beetles from Pinus thunbergii and Pinus massoniana infected by pine wood nematode” (ID: forests-1432734) responded to my previous comments. I appreciate their effort in revising their work. I find most of the responses convincing. However, I still think that this kind of study should contain analyses of Bursaphelenchus uninfected tree hosts, as without such data it is not possible to understand how infection affect dendrophilic beetle communities, especially as according to this study, no beetle species (except single longhorn beetle) was vector of this nematode. So, it is unclear if beetle communities change because of the presence of nematodes having impact on the host plant (of both – nematodes and beetles) or if there are some other factors shaping beetle communities. Moreover, I disagree that this study examines temporal changes of beetle communities. To do this, the same host tree should be investigated from health until death, which is basically impossible. This study compares beetle communities from trees of different stages caused by nematode infection. I suggest being specific and avoiding the term “temporal”.

Response: Most pine trees susceptible to PWD are weak trees with declining tree vigor, weak trees are closely related to pine wood nematode. Therefore, the main purpose of this experiment is to investigate the species and distribution of xylophagous beetles in pine trees at different infected stages. On the basis of previous studies, we have a detailed understanding of the species, quantity and distribution position of xylophagous beetles species and richnesson P. massoniana and Pinus thunbergii in China. Song found that by comparing the xylophagous beetles of healthy pine trees with those of infected pine trees, there were few species and quantities of xylophagous beetles on healthy pine trees. There is only one species of Tomicus piniperda on healthy pines, but there are more than 20 kinds of xylophagous beetles on infected pines (Song, J.Y.; Luo, Y.Q.; Shi, J.; Yan, X.; Chen, W.; Ping, J. Niche of insect borers within Pinus massoniana infected by pine wood nematode. Frontiers of Forestry in China. 2006, 1, 4, 460–463). This indicates that the colonization of pine wood nematodes has greatly changed the community of xylophagous beetles, and we should pay more attention to the species and distribution of xylophagous beetles in pine trees after the colonization of pine wood nematode. We are concerned about the changes of species and quantity of xylophagous beetles after infection with pine wood nematode. Therefore, we did not set up a control group of uninfected pines. We hope that through further analysis, To determine the function of insects living on the weak wood of P. massoniana and Pinus thunbergii in the pine forest ecosystem and the interspecific relationship between xylophagous beetles. In conclusion, pine trees that are not infected with pine wood nematode disease are not the focus of this experiment (line 67-69 in the revised version). 

For “temporal”, we modified the expression in the manuscript to avoid its use. The full manuscript was modified.

SPECIFIC COMMENTS:

Comment 1: Minor comment. As far as I know, in the current taxonomy of beetles Scolytinae is a subfamily, not family, so you should not simply change the scientific name of this taxon.
Response: We describe the classification of bark beetle in more detail. We modified this sentence to “These species included six species in the family Curculionidae, subfamily Scolytinae, Ipidae, including Xyleborus hawaiiensis (Perkins, 1900), Orthotomicus laricis (Fabricius, 1792), Orthotomicus erosus (Nordl, 1888), Pityogenes trepanatus (Nördlinger, 1848), Pityokteines spinidens (Reitt, 1894), and Cyrtogenius luteus (Blandford, 1894), accounting for 53.70%”. The families, subfamilies and families to which bark beetle belong are described in detail (line 188 in the revised version).

This manuscript is a resubmission of an earlier submission. The following is a list of the peer review reports and author responses from that submission.

Round 1

Reviewer 1 Report

The paper by Chu et al. investigated the distribution of wood-boring beetles on two Pinus spp. affected by the PWN. Despite the topic is potentially interesting given the interest on the Pine Wood Nematode worldwide, the quality of the paper is not sufficient for publication. The main flaws are:

1) English. the paper is badly written and it is very difficult for the reader to understand what the authors did.

2) the lack of details in the Materials and Methods, which does not allow the reader to understand whether the sampling design and the methodology used are correct.

3) Number of replicates: it is not clear whether the authors felled three trees per each stress category per each Pinus spp. or only one. In both cases, the number of replicates is too low (see also point no. 4).

4) Stress stages: the authors divided the analyzed tree in three categories, i.e., weak wood, dying wood and withered tree. This classification is a key aspect of the whole study but it is really difficult to understand whether it is reliable or not. In addition, the authors state that they selected trees infected by the PWN, but they did not explain how it is possible to distinguish trees really stressed by the PWN or by other biotic or abiotic factors. In addition, given the high variability that there can be among trees possibly included in each category (e.g., in term of diameter, height, oldness), three replicates are definitely not sufficient.

3) References are often not adequate and correct. I found several sentences associated with references that are not proper for the statement, and I thus recommend the authors to carefully recheck all of them.  

SPECIFIC COMMENTS:

Line 46: delete “were”

Line 48: define what “mu” means

Line 49: delete “strains”

Line 51: “add a , after respectively”

Line 56-57: awkward sentence. I suggest to change as “depends on insect vectors and human-driven spread”

Line 57: I suggest to modify as “The main vectors are the Monochamus longhorn beetles (Coleoptera; Cerambycidae), but about 45 other woodboring beetles, including (…list here the non-Monochamus spp….), can potentially carry the PWN.” In addition, I think that citations for this statement are not the best ones. I would suggest to use the following one: “Robertson et al. (2008) Potential insect vectors of Bursaphelenchus spp.(Nematoda: Parasitaphelenchidae) in Spanish pine forests.” 

Line 62-63: delete the last sentence of the paragraph as it is a repetition.  

Line 77: what do you mean with invasion? Do you mean colonization?

Line 101-108: you need to provide more info band details here, otherwise it is difficult to understand how reliable this classification is. This represents a crucial aspect of this study. Including pictures as Supplementary Materials would help the reader to better understand the different categories.

Line 107-108: do you mean three trees for each category for each sampling site?

Line 110: what do you

Lime 114: what do you mean with “invasive parts”? How did you dissect the tree?

Line 160: why did you retain only 16 out of the 20 species for analysis?

Line 161: what does it mean that you retained 9 species for each forest type?

Reviewer 2 Report

I have provided an annotated PDF with many comments for the authors to consider.  I am particularly worried about the experimental design.  The authors either did not account for size variation across treatments and replicates or provided no information on this.  For an experiment like this, when using multiple trees that undoubtedly varied in diameter at sampled sections as well as total tree height, you have to account for this variation by standardizing data.  I saw no evidence that this was completed.

The term woodboring beetle is used throughout.  However, several of the beetles mentioned are not woodboring beetles and instead inhabit only the phloem (bark beetles) or are predaceous.  I would recommend really thinking through your species list and breaking out the species into ecological roles.  You can then decide to focus only on the woodborers, the woodborers and bark beetles, or everything.  In my view, I would not have predators and other such species in the analyses and just focus on woodborers (in this case, only cerambycids) and bark beetles.

The statistical methods need to be better described along with all the community metrics you tested.  I would also consider using rarefaction as you have uneven sampled area as is.  Rarefaction will help account for this.

The discussion needs a lot of work and much of it is outside the scope of this work.  It would benefit from a heavy editing and a reconsideration of what is important in terms of data generated during this study.

Reviewer 3 Report

I revised the manuscript entitled “Diversity and distribution of woodboring beetles from Pinus  thunbergii and Pinus massoniana infected by pine wood nematode”.

This manuscript have an exciting aim and a novel focus of study of pine wilt disease out of the classic pair nematode-insect vector.

Unfortunately, I think that the manuscript is not suitable to publishing in the present stage. Please, taken into account the Major Considerations that I detail below:

MC1: The main trouble of this work is the absence of negative control in the experimental design. Authors consider three replicates of infectec Pinus thumbergii and other three replicates of infected P. massoniana, but no non-infected trees as control were considered. Authors aim define the communities of infected trees of both species, but, what is the communities of non-infected? Are different or equal? Without negative controls the results obtained with the present experimental design were incomplete.

MC2: Author base their objectives in increase the knowledge of other potential vectors of PWN in the woodboring community, citing in line 58 that “45 woodboring insects [species, I suppose that authors refers] also have an ability to carry PWN” , accordingly three references. At the present, I know that PWN is exclusively vectored by Monochamus genus. The referred literature refer the ability of other non Monochamus species to carry out other species of Bursaphelenchus nematode, but not the PWN B. xylophilus. So authors need clarify it for avoid confusions in several parts of the text.

MC3: Conceptual considerations about ecological niche. The ecological niche is the hyperspace of variables that define the where (into these hyperspace) a species could develop their life cycle. To know the entire number of variables that compose this space is a hard and never complete work.  The formulas of niche breadth and overlap consider “n” as the number of niche resource levels (=level of variables). Unfortunately, authors do not explain how calculate this “n” number. What variables or resources has been considered for the authors? I don’t know it.  I only could deduce that author use the variable “tree part”, but I can’t understand the other variables. I think that authors need improve the explanation of the this section, avoiding the mistake between “ecological niche” and “biotope”. All discussion of niche bread need be focused under this perspective, plus consideration of other species interactions, not only competence, as intraguild predation.

MC3. About references: I consider that some references do not justified some sentences of the text. I.E: Reading the line 122, I hope that the reference 26 should be related to a key or similar for identified the woodboring species, but is the title of this paper is “Precise identification of different stages of a tick, Ixodes granulatus Supino, 1897 (Acari: Ixodidae).” But, authors do not include Aracnida in the considered taxa.

Also, authors need consider the following minor considerations (mc)

mc1.- The text need a deeply edition process. Several errors are due to send the manuscript before a calm revision, without hurry. I.E:

               First paragraph (lines 35 to 41), need be removed, is a part of the editorial template.

               Line 52: “pine forests” instead “pine forest”

               Paragraph of lines 92 to 99: Wrote in italic.

               Line 142: remove the term “Shannon diversity index”, because was not referred in the results. Authors only use the Simpson diversity index.

               All formulas: Remove the terms “(reference)”

               Formula of line 152: Remove it because is not cited in results and discussion.

               Figure 1: Is very difficult to read this figure with the present representation of the phylogenetic tree. I think that this figure need be improved using a more classic representation and increasing the font size. On the other hand, what means the grey points?

               Figures 2 and 3: I doubt about what stage of infection corresponds (initial, medium or terminal) these figures? They are the consensus? These figures are very nice and informative, congratulations, but need this improvement.

               Figures 2 and 3 are not referred in the text.

               Line 201: “P. thumbergii” is wroted, and the first words of the following line indicates “P. massoniana”.

               Figure 4: Increase the font size of the x axe.

               Line 234: Referring Figure 5 as graphic representation of Simpson index, but the Y axe of this figure are labeled as “Species richness” instead “Simpson index”. According with the aesthetical of the other figures, I think that Figures 5 and 6 need be printed in color.

               Figure 7: need improving their explanation in the figure caption, in the present stage is very difficult to understand.

Line 305: “Blatt et al”  instead  “Blatt ” alone.

Line 308: “Vicente” instead “Vicent”

Line 334: What means “temporal niche”?

Table S3: Remember that the value of the Simpson index is inverse to the diversity value, that is, low values of the index correspond to high values of diversity. Usually, the inverse of Simson index or Gini-Simpson index are used to improve their compression. What index has been used? Please, clarify it.

Line 433: the reference includes as authors a part of their affiliations:  “Unit, A.; Centre, I.D.; Medical, I.F.; Systematic, C.F.I.; Sciences, 433 E.N.R.”